Blood glucose, insulin and glycogen profiles in Sprague-Dawley rats co-infected with Plasmodium berghei ANKA and Trichinella zimbabwensis

Murambiwa Pretty 1
Nkemzi Achasih Quinta 1
Mukaratirwa Samson mukaratirwa@ukzn.ac.za 1 2
1 School of Life Sciences, University of KwaZulu-Natal , Durban , South Africa
2 One Health Center for Zoonoses and Tropical Veterinary Medicine, Ross University School of Veterinary Medicine , Bassterre , Saint Kitts and Nevis
Braga Erika
Electronic publication date: 2022 Jul 29
Publication date: 2022
Volume: 10
Electronic Location ID: e13713
Received 2022 Mar 2; Accepted 2022 Jun 21
Copyright: ©2022 Murambiwa et al.
Copyright year: 2022
Copyright holder: Murambiwa et al.
License: This is an open access article distributed under the terms of the Creative Commons Attribution License, which permits unrestricted use, distribution, reproduction and adaptation in any medium and for any purpose provided that it is properly attributed. For attribution, the original author(s), title, publication source (PeerJ) and either DOI or URL of the article must be cited.
License URL: https://creativecommons.org/licenses/by/4.0/

Keywords: Co-infection, Blood glucose, Insulin, Liver glycogen, Muscle glycogen, Plasmodium berghei, Trichinella zimbabwensis

Funding: University of KwaZulu-Natal National Research Foundation of South Africa This work was supported by incentive funding for research awarded to Samson Mukaratirwa by the University of KwaZulu-Natal. Pretty Murambiwa received funding from the National Research Foundation of South Africa. The funders had no role in study design, data collection and analysis, decision to publish, or preparation of the manuscript.

==============================
Background

Plasmodium falciparum and tissue dwelling helminth parasites are endemic in sub-Saharan Africa (SSA). The geographical overlap in co-infection is a common phenomenon. However, there is continued paucity of information on how the co-infection influence the blood glucose and insulin profiles in the infected host. Animal models are ideal to elucidate effects of co-infection on disease outcomes and hence, blood glucose, insulin and glycogen profiles were assessed in Sprague-Dawley rats co-infected with P. berghei ANKA (Pb) and Trichinella zimbabwensis (Tz), a tissue-dwelling nematode.

Methods

One-hundred-and-sixty-eight male Sprague-Dawley rats (weight range 90–150 g) were randomly divided into four separate experimental groups: Control (n = 42), Pb-infected (n = 42), Tz-infected (n = 42) and Pb- + Tz-infected group (n = 42). Measurement of Pb parasitaemia was done daily throughout the experimental study period for the Pb and the Pb + Tz group. Blood glucose was recorded every third day in all experimental groups throughout the experimental study period. Liver and skeletal muscle samples were harvested, snap frozen for determination of glycogen concentration.

Results

Results showed that Tz mono-infection and Tz + Pb co-infection did not have blood glucose lowering effect in the host as expected. This points to other possible mechanisms through which tissue-dwelling parasites up-regulate the glucose store without decreasing the blood glucose concentration as exhibited by the absence of hypoglycaemia in Tz + Pb co-infection group. Hypoinsulinemia and an increase in liver glycogen content was observed in Tz mono-infection and Tz + Pb co-infection groups of which the triggering mechanism remains unclear.

Conclusions

To get more insights into how glucose, insulin and glycogen profiles are affected during plasmodium-helminths co-infections, further studies are recommended where other tissue-dwelling helminths such as Taenia taeniformis which has strobilocercus as the metacestode in the liver to mimic infections such as hydatid disease in humans are used.

Introduction

Co-infection and poly-parasitism remain a major health challenge in sub-Saharan Africa (SSA) (Onkoba, Chimbari & Mukaratirwa, 2015). There is a geographical overlap in the endemicity of various human parasitic diseases such as malaria and helminthiasis in SSA (Onkoba, Chimbari & Mukaratirwa, 2015) which include tissue-dwelling helminth (TDH) parasites (Onkoba, Chimbari & Mukaratirwa, 2015), and 90% of all global malaria related deaths also occur on SSA (World Health Organization, 2019). Undoubtedly, malaria-helminth co-infections are a common phenomenon in SSA (Onkoba, Chimbari & Mukaratirwa, 2015). Ascaris lumbricoides, Taenia solium cysts, Echinococcus spp cysts, filarial worms, Schistosoma spp, Fasciola spp and Trichinella spp are some of the common TDHs reported worldwide and have either their larvae or adults stages passing through host tissues other than the gastrointestinal tract and are common in SSA (Onkoba, Chimbari & Mukaratirwa, 2015).

Glucose homeostasis during Plasmodium spp mono-infection without considering co-infection with helminth parasites is well understood and documented under laboratory conditions. However, there is paucity of information on glucose homeostasis during Plasmodium spp and TDHs co-infection. Trichinella zimbabwensis (Tz) has been successfully used in laboratory animal model experiments for TDHs (Onkoba, Chimbari & Mukaratirwa, 2015) and in determining chemokine, cytokine and haematological profiles in Sprague-Dawley rats co-infected with Plasmodium berghei ANKA (Murambiwa et al., 2020) and the authors propose extending the use of this parasite to determine the blood glucose, insulin and glycogen profiles during co-infection with P. berghei ANKA (Pb) to fully understand the pathophysiological implications of co-infection.

During experimental Trichinella spp mono-infection, it has been reported that trichinella-induced hypoglycaemia could be mediated via three possible mechanisms, viz; high glucose consumption by trichinella parasites, reduction in the absorptive capacity of the intestine or impairment of glucose production by the liver (Wu et al., 2009). Increase of glucose uptake in infected muscle cells of the host has also been reported as a trichinella-induced hypoglycemia mechanism (Wu et al., 2009). Trichinella-induced hypoglycaemia in humans, mice and dogs has also been reported in other studies (Nishina, Suzuki & Matsushita, 2004; Reina et al., 1989; Steward, 1983; Montgomery, Augostini & Steward, 2003). During trichinella infection, there is increased demand for glucose by both trichinella muscle larvae and infected muscle cells for their rapid growth and metabolism (Wu et al., 2009). However, in the Nile crocodile (Crocodylus niloticus) experimentally infected with T. zimbabwensis no hypoglycaemia was detected in infected groups contrary to previous findings in mammals (La Grange & Mukaratirwa, 2014). In high infection groups, peak blood glucose concentration was recorded 35–49 days post infection (La Grange & Mukaratirwa, 2014). A significant increase in blood glucose concentration 30 days post infection was also reported in pigs (Oltean et al., 2012) which coincided with trichinella larvae being established in the striated muscles of host.

An increase in glycogen concentration was observed at day 8 post-infection and day 18 post-infection in mice infected with T. spiralis and T. pseudospiralis infections (Wu et al., 2009). However, at day 28 and 48 post-infection with T. spiralis and T. pseudospiralis infections, the study showed depletion of glycogen stores (Wu et al., 2009). Increased glucose uptake through insulin signaling pathways has been speculated to be the possible mechanism of increased glycogen accumulation post-infection (Wu et al., 2009). Infection of mice with T. spiralis and T. pseudospiralis has been reported to cause an initial decrease in insulin concentration, which is restored back to normal (Wu et al., 2009). Increased insulin concentration in mice infected with T. zimbabwensis compared to the control has also been reported (Onkoba et al., 2016). However, the effect of Pb + Tz co-infection on insulin, liver and muscle glycogen concentration remains obscure. We recently reported that Tz infection predisposed the co-infected animals towards rapid development of Pb parasitaemia during co-infection (Murambiwa et al., 2020). However, there is still a paucity of information on how this phenomenon influences glucose homeostasis, insulin and glycogen levels during co-infection.

Currently, glucose homeostasis during mono-infection of the two parasites is fairly well understood and documented, however, glucose homeostasis during co-infection, remains obscure. It is against this background that the authors attempt to use an animal model aimed to determine the blood glucose, insulin profiles and glycogen concentration in Sprague-Dawley rats co-infected with Pb ANKA and Tz and the results will give an insight of the impact co-infection may have in the epidemiology of malaria in geographical areas where the two are known to overlap.

Materials & Methods

Study animals and study design

Male Sprague Dawley rats (90–150 g) used in this study were bred and maintained at the Biomedical Research of the University of KwaZulu-Natal, South Africa (Murambiwa et al., 2020). The animals were fed rat chow (Meadow feeds, Pietermaritzburg, South Africa) with free access to water and maintained following standard laboratory conditions (Murambiwa et al., 2020). The animals were randomly divided into control (n = 42), T. zimbabwensis-infected (Tz) (n = 42), Plasmodium berghei-infected (Pb) (n = 42) and Pb + Tz-infected (n = 42) groups using a simple random method (Murambiwa et al., 2020) with the only variation in the parameters which were measured in this study. Severe weight loss, anaemia and body temperature <32 °C were considered as an endpoint for Pb- and Pb + Tz infected groups.

At the end of the experimental period, animals were humanely sacrificed using CO2 on day 0, 7, 14, 28, 35 and 42 (n = 6 in each group) post Tz-infection for collection of sera and rat carcasses for determination of serum insulin and trichinella adult worm and muscle larvae load respectively (Murambiwa et al., 2020). Liver and skeletal muscle were harvested, snap frozen and stored at −70 °C for determination of glycogen concentration. ARRIVE Guidelines for reporting animal research were followed in this experiment as reported by Kilkenny et al. (2010). The completed ARRIVE Guidelines checklist is found in the Supplemental Files.

Ethical statement

Experimental procedures and protocols for the study were reviewed and approved by the University of KwaZulu-Natal Animal Ethics Committee (AREC/018/016).

Inoculation and determination of Pb and Tz infection

The procedures followed for the induction and determination of Pb in Sprague Dawley rats are as described by Murambiwa et al. (2020). The Pb ANKA strain, donated by the University of Cape Town, South Africa, was used for this study and propagated in Sprague-Dawley rats before being frozen at −80 °C in liquid nitrogen. The parasite was propagated in stock rats as described by Ademola & Odeniran (2016) with each experimental rat receiving 105 parasitized red blood cells (RBCs) intraperitoneally and control animals were given phosphate buffered saline vehicle in equal volume via the same route. Parasitaemia was determined daily after Pb infection throughout the duration of the experiment as described by Murambiwa et al. (2020).

Description of the crocodile-derived Tz parasite strain and the procedures followed in the induction and inoculation in Sprague Dawley rats have already been described by Murambiwa et al. (2020). First-stage larvae of Tz was administered to the experimental animals per os at a dose of 3 mL/g of rat body weight. Harvesting of adult worms from the intestines were as described by Mukaratirwa et al. (2003) and the protocol for digestion of muscle tissue for recuperation of larvae was followed as described previously by Kapel & Gamble (2000). Determination of Tz adult worms in the intestines was done at day 7, 14 and 21 post-infection and muscle larvae at day 28, 35 and 42 post-infection (Murambiwa et al., 2020).

Determination of serum insulin

An ultrasensitive rat insulin ELISA kit (DRG Instruments GmBH, Marburg, Germany) was used for the determination of serum insulin. This immunoassay is a quantitative method that utilizes two monoclonal antibodies specific for insulin and procedures were followed as described by the manufacturer. The lower limit of detection was 1.74 pmol/L.

Determination of liver and gastrocnemius muscle glycogen concentration

Glycogen was determined as described by Ngubane, Masola & Musabayane (2011). Homogenization of pre-weighed (0.25−0.5 g) liver and gastrocnemius muscle tissue samples was done in two mL of 30% potassium hydroxide (300 g/L) and boiled at 100 °C for 30 min and then cooled in ice-saturated sodium sulfate. Ethanol was used to precipitate glycogen, which was then pelleted and resolubilised in deionized water. Treatment with anthrone reagent was done for glycogen content determination, which was done at 540 nm using a Spectrostar Nano microplate reader (BMG Labtech, Ortenberg, Germany).

Data analysis

Comparison of the differences between the experimental groups (median and the 25% −75% quartiles) was done using GraphPad Instat Software, Version 4.00, San Diego, CA, USA. Effects of co-infection on blood glucose, serum insulin as well as liver and gastrocnemius muscle glycogen concentration were determined using one-way analysis of variance (ANOVA), followed by Bonferroni post hoc test. A value of p < 0.05 was considered statistically significant.

Results

There were no adverse events in this experimental protocol.

Effect of co-infection on Pb parasitaemia and Tz adult worms and muscle larvae load

The effect of coinfection on Pb parasitaemia and Tz adult worms and muscle larvae load has been reported elsewhere by Murambiwa et al. (2020) (Figs. 1 and 2) where it is shown that percentage parasitaemia of the Pb mono-infected group was generally lower than in the co-infected group (Fig. 1) .

Adult worm counts were recuperated in both experimental groups at day 7 and 14 post-infection and no adult worms were recovered as from day 21 post infection while muscle larvae (ML) counts were detected from day 28 post-infection and relatively higher in the co-infected group (Fig. 2).

Effects of Tz and Pb co-infection on blood glucose concentration

Effects of Tz and Pb co-infection on blood glucose concentration is shown in Fig. 3. Plasmodium berghei (Pb) mono-infected group had a significant reduction in blood glucose concentration at day 7 post Pb infection (P < 0.05) in comparison to control group. Also, at day 14 post Pb infection, there was a further reduction in blood glucose concentration in Pb mono-infected group (P < 0.001) in comparison to control group. Blood glucose concentration in Pb mono-infected group was significantly reduced, in comparison to Tz mono-infected (P < 0.001) and co-infected (P < 0.001) experimental groups at day 14 post Pb infection. There were no significant differences in blood glucose concentration among experimental groups at day 21 post Pb infection.

Figure 1 Percentage parasitaemia in male Sprague-Dawley rats infected with Plasmodium berghei (Pb) only and co-infected with Pb and Trichinella zimbabwensis (Pb + Tz) (Murambiwa et al., 2020; CC-BY-NC-ND).

Day 0 represents the day of Pb infection when Tz muscle larvae were in the rat muscle at day 28 post Tz infection. Values are presented as means and vertical bars indicate standard error of mean (SEM). N = 6 for each group. **P < 0.01, ***P < 0.001 (https://creativecommons.org/licenses/by-nc-nd/2.0/legalcode).

Figure 2 Mean number of intestinal adult worms (AW) and muscle larvae counts (ML)/gram of muscle (lpg) recovered from rats infected with Trichinella zimbabwensis (Tz) and the group co-infected at day 28 post-infection (Murambiwa et al., 2020; CC-BY-NC-ND).

Values are presented as means and vertical bars indicate standard error of mean (SEM). N = 6 for each group (https://creativecommons.org/licenses/by-nc-nd/2.0/legalcode).

Figure 3 Comparison of the effects of Plasmodium berghei (Pb) and Trichinella zimbabwensis (Tz) mono-infection and co-infection (Pb + Tz) on blood glucose concentration.

Day 0 represents the day of Pb infection when Tz muscle larvae were in the rat muscle at day 28 post Tz infection. Values are presented as means and vertical bars indicate standard error of mean (SEM). N = 6 for each group. **P < 0.01, ***P < 0.001.

Effects of Tz and Pb co-infection on serum insulin concentration

There was a significant reduction in serum insulin concentration in Tz mono-infected (P < 0.05) and co-infected (P < 0.05) experimental groups at day 0 post Pb infection in comparison to both the control and Pb infected groups (Table 1). The co-infected group serum insulin concentration was significantly reduced (P < 0.05) at day 14 Pb infection in comparison to the control group. There were no significant differences in serum insulin concentration of all experimental groups.

Table 1 Median values (25%–75% quartiles) of insulin, liver and muscle glycogen in Sprague–Dawley rats co-infected with Plasmodium berghei ANKA (Pb) and Trichinella zimbabwensis (Tz) (Pb + Tz), Pb mono-infection and Tz mono-infection.

Parameter	Days post P. berghei infection	Experimental groups	
		Control	Pb	Tz	Pb + Tz	
Insulin (pmol/l)	Day 0	21.93 (19.70–46.89)	21.93 (19.70–46.89)	2.69 (2.48–4.17)	2.69 (2.48–4.17)	
Day 7	21.93 (19.70–46.89)	25.66 (8.75–39.96)	4.35 (2.26–25.84)	5.39 (3.04–12.92)	
Day 14	21.93 (19.64–46.89	15.93 (9.37–25.71)	27.06 (15.97–40.54)	18.62 (14.14–26.97)	
Liver glycogen (mg/g tissue)	Day 0	0.73 (0.52–0.77)	0.57 (0.49–0.50)	0.64 (0.45–0.75)	0.68 (0.41–0.79	
Day 7	0.62 (0.53–0.75)	0.65 (0.54–0.72)	0.28 (0.20–0.33)***	0.37 (0.28–0.39)***	
Day 14	0.63 (0.43–0.76)	0.60 (0.49–0.75	0.58 (0.47–0.62)	0.35 (0.25–0.53)*	
Muscle glycogen (mg/g tissue)	Day 0	–	–	0.11 (0.10–0.13	0.14 (0.09–0.16)	
Day 7	–	–	0.18 (0.12–0.30)	0.25 (0.17–0.30)	
Day 14	–	–	0.18 (0.18–0.22)	0.27 (0.21–0.36)	
Notes.

Day 0 represents the day of Pb infection when Tz larvae were now established in the rat muscle at day 28 post Tz infection. N = 6 for each group.

Figures with asterisks superscript within a row under experimental groups are significantly different from the control group (* = P < 0.05, ** = P < 0.01, *** = P < 0.001).

Effects of Tz and Pb co-infection on liver and gastrocnemius muscle glycogen concentration

There was a significant reduction in liver glycogen concentration in Tz mono-infected (P < 0.001) and co-infected (P < 0.001) experimental groups at day 7 post Pb infection in comparison to the control group (Table 1). A significant reduction in liver glycogen concentration in Tz mono-infected (P < 0.001) and co-infected (P < 0.01) experimental groups was also observed at day 7 post Pb infection in comparison to Pb mono-infected group (Table 1). Additionally, liver glycogen concentration in co-infected group was significantly reduced (P < 0.05) at day 21 post Pb infection in comparison to the control group. There were no significant differences in liver glycogen concentration among all experimental groups at day 0 post Pb infection.

There was a gradual increase in muscle glycogen concentration at day 0, 7 and 14 post Pb infection although the differences among groups were not statistically significant (Table 1). Co-infected group had elevated muscle glycogen concentration at day 7 and 14 post Pb infection in comparison to Tz mono-infected group, although the differences were not statistically significant.

Discussion

Elevated Pb parasitaemia in Pb mono-infected group coincided with reduced blood glucose concentration in comparison to control group at day 7 post Pb infection. Blood glucose concentration in the Pb mono-infected group was significantly reduced in comparison to control group at day 7 post Pb infection and in comparison, to all other experimental groups at day 21 post Pb infection. Indeed, blood glucose lowering effects of Plasmodium parasites have been previously reported (Mehta, Sonawat & Sharma, 2005). Malaria parasites have been reported to precipitate hypoglycaemia through unclear mechanisms (English et al., 1998; Thien, Kager & Sauerwein, 2006). There are studies which have reported that hypoglycemia is induced through utilization of glucose and gluconeogenic substrates such as thiamine in an effort to meet the parasite’s increased energy demands (Krishna et al., 1999).

During P. falciparum infection, studies have reported depressed aerobic glycolysis and increased acidosis due to lactic acid in the host following parasite-induced depletion of vital gluconeogenic substrates such as thiamine (Krishna et al., 1999). Furthermore, other studies have reported that malaria parasites are completely dependent on the host for all energy requirements (Phillips, 1989). On the other hand, the liver plays a major role in glucose homeostasis and induced hypoglycaemia may also be due to P. falciparum induced hepatocellular damage (Dekker et al., 1997) which causes loss of intracellular fluid components form the hepatocytes such as liver enzymes to the extracellular fluid compartment (Dekker et al., 1996; Dekker et al., 1997; Kauser et al., 2010). Hepatocellular damage may also lead to slow insulin receptor recycling (Onyesom & Agho, 2011), thereby aggravating malaria induced hypoglycaemia.

Increased glucose demands during trichinella infection also coincide with NBL larvae migration, penetration, establishment and encystment in the striated muscle nurse cells (Wu et al., 2009). A temporary decrease in blood glucose concentration between day 8 and 28 post T. spiralis and T. pseudospiralis infection, coinciding with peak larval growth and development has been reported by Wu et al. (2009). The same study further reported increased glycogen accumulation in infected muscle cells during transient hypoglycemia phase (Wu et al., 2009). Indeed, it is speculated that trichinella parasite may regulate glycogen synthesis in the infected muscle cells in line with its glucose requirements for its growth and metabolism (Wu et al., 2009).

Hyperinsulinemia was observed in the current study and results are in agreement with Balaji, Deshmukh & Trivedi (2020) who reported this phenomenon in a case cerebral malaria or severe malarial anaemia in children and Wu et al. (2009) have reported that the transient hypoglycemia reported between 8 and 28 days post T. spiralis and T. pseudospiralis infection was a result of increased glucose uptake by infected muscle cells mediated through up regulation of insulin signaling pathway related genes, and not necessarily through increased blood insulin concentration. However, there were no statistical differences in insulin concentration of all the experimental groups in the current study, 14 days post P. berghei infection. This could be suggestive of differences in blood glucose uptake mechanisms following nurse cells formation completion. In comparison to control group, reduction in serum insulin concentration in Pb mono-infected group persisted at day 7 post Pb infection in comparison control group.

Glucose transport is insulin mediated, via specific insulin signaling pathways in both the skeletal muscle and adipose tissue (Wu et al., 2009). Insulin mediated glucose transport has also been reported to enhance conversion of glucose to glycogen storage molecule (Wu et al., 2009) and the same study alsoStudies have also shown that infection of mice with T. spiralis and T. pseudospiralis causes an initial decrease in insulin concentration, which is restored back to normal (Wu et al., 2009). Interestingly, increased insulin concentration in mice infected with T. zimbabwensis compared to the control has also been reported (Onkoba et al., 2016). Also, studies have shown a difference in glucose metabolism handling between children and adults in falciparum malaria through unclear mechanisms (Dekker et al., 1996; Balaji, Deshmukh & Trivedi, 2020). In the current study, weanling male Sprague-Dawley rats (90–150 g) were used for all experimental protocols.

There was elevated liver glycogen concentration which coincided with reduction of blood glucose and plasma insulin through unclear mechanisms. Significantly reduced insulin concentration in both Tz mono-infected and co-infected experimental groups in comparison to both the control and Pb infected groups coincided with reduced muscle glycogen concentration at day 0 days post Pb infection. Gradual elevation of insulin concentration in both Tz mono-infected and co-infected experimental groups coincided with a concomitant gradual elevation of muscle glycogen concentration in both Tz mono-infected and co-infected experimental groups at day 7 and 14 post Pb infection. Additionally, elevated Tz infection coincided with elevated muscle glycogen concentration. Although these differences did not reach statistical significance, they could be of biological importance. Interestingly, there was a significant reduction in glycogen concentration in Tz mono-infected and co-infected groups in comparison to both control and Pb mono-infected group. Indeed, Tz infection could cause hypoglycemia through three possible mechanisms; high glucose consumption by developing stages of Trichinella parasites, reduction in the absorptive capacity of the intestine or impairment of glucose production by the liver (Wu et al., 2009). Nishina, Suzuki & Matsushita (2004) has previously reported hypoglycemia through unknown mechanisms in mice. Studies have also reported hypoglycemia in dogs experimentally infected with T. spiralis (Reina et al., 1989). Several Trichinella induced hypoglycemia mechanisms have been postulated, such as increased consumption of glucose by rapidly growing Trichinella larvae within muscle cells (Wu et al., 2009). Steward (1983) reported that trichinella induced hypoglycemia was mediated via increased total glycogen in infected muscle tissue as well as increased glycogen content in trichinella larvae. Other researchers from one study have ascribed the trichinella induced hypoglycemia to increased metabolic activity associated with sugar metabolism of nurse cells (Montgomery, Augostini & Steward, 2003).

Interestingly, there were no significant differences in the gastrocnemius muscle glycogen concentration of the experimental groups of animals throughout the experimental period. However, the observed increase in gastrocnemius muscle glycogen concentration of the co-infected group at day 7 and 14 post Pb infection could be of biological importance since glucose is transported in the muscle via insulin mediated GLUT 4 transporters (Azpiazu et al., 2000; Ferrer et al., 2003).

Conclusions

To the best of authors’ knowledge, there is paucity of studies that have investigated the effects of Pb and Tz co-infection on blood glucose, glycogen concentration and insulin profiles in male Sprague-Dawley rats as an animal model for human infection. The authors conclude from our study that Tz mono-infection and Tz + Pb co-infection did not have blood glucose lowering effect in the host. There was hypoinsulinemia and increase in liver glycogen content in Tz mono-infection and Tz + Pb co-infection groups but the triggering mechanism remains unclear. They also postulate that there are other possible mechanisms through which the tissue-dwelling parasite up-regulates the glucose store without decreasing the blood glucose concentration as exhibited by the absence of hypoglycaemia in Tz + Pb co-infection group. The limitations of the study include the use of T. zimbabwensis which has striated muscles as site of predilection and future studies should include other tissue-dwelling helminths such as Taenia taeniformis which has strobilocercus as the metacestode in the liver to mimic infections such as hydatid disease in humans.

Supplemental Information

Supplemental Information 1 Raw data for glucose and liver glycogen

Click here for additional data file.

Supplemental Information 2 Author checklist form

Click here for additional data file.

Supplemental Information 3 Summary statistics of comparison between experimental groups

Click here for additional data file.

We acknowledge the assistance rendered by staff from the Biomedical Research Unit and the Parasitology Laboratory of the University of KwaZulu-Natal, Westville Campus, in looking after the experimental animals, and processing and analyzing the samples.

Additional Information and Declarations

Competing Interests

Author Contributions

Animal Ethics

Data Availability

The authors declare there are no competing interests.

Pretty Murambiwa conceived and designed the experiments, performed the experiments, analyzed the data, prepared figures and/or tables, authored or reviewed drafts of the article, and approved the final draft.

Achasih Quinta Nkemzi performed the experiments, analyzed the data, prepared figures and/or tables, and approved the final draft.

Samson Mukaratirwa conceived and designed the experiments, performed the experiments, analyzed the data, prepared figures and/or tables, authored or reviewed drafts of the article, and approved the final draft.

The following information was supplied relating to ethical approvals (i.e., approving body and any reference numbers):

Experimental procedures and protocols for the study were reviewed and approved by the University of KwaZulu-Natal Animal Ethics Committee (AREC/018/016).

The following information was supplied regarding data availability:

The raw data are available in the Supplemental Files.

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
