# Peer review of "Blood glucose, insulin and glycogen profiles in Sprague-Dawley rats co-infected with Plasmodium berghei ANKA and Trichinella zimbabwensis"

_PeerJ, doi:10.7717/peerj.13713_

## Round 0.1 · original submission · Major Revisions

The review process is now complete, and two reviews (including annotated manuscript) are included at the bottom of this letter. All reviewers and I agree that your manuscript deserves to be published. Although there is considerable merit in your paper, we also identified some concerns that must be considered in your resubmission.

1-Please, consider reviewing the abstract aiming to emphasize the importance and novelty of the study. Include a conclusion and delete excessive details related to the methods and results.

2- Figures 1 and 2 have been published already. Please, describe them only as a background, not as a new and original result.

Please, provide, point-to-point responses according to the comments made by the Reviewers in the new version of your manuscript.

·

Basic reporting

1. The problem statement was not stated clearly, hence not explicit in the intro section.
2. Some literature references were not properly cited across the text, some cited references were missing in the ref. list, while often, most of the cited references were too old. It is understandable that there are not too many literature on Tz, but other spp of Trichinella or tissue-dwelling nematodes could be cited.

Experimental design

Some vital information about inoculation of the parasites and animal handling & sacrifice were not provided. The author should provide detailed info about these.

Validity of the findings

1. Any evidence of permission by the journal or publisher of the Fig. 1&2 to be for publication in PeerJ?
2. The use of 2-way ANOVA for table 1. was not appropriate. Also, posthoc test should have been adopted to do the ranking for differences in means.

Additional comments

1. The Title is too long. Suggestion has been provided in the text
2. Footnote should be provided for the table
3. There should be consistency in the references list.

·

Basic reporting

The article is well written and meets the standard presentation of this journal.
Authors introduced their subject with a clear research question referring to relevant literature to give the reader context. Raw data was supplied as supplementary files and due process was taken to give recognition to authors whose findings are available already in literature to complete the methodology used in this paper. Unfortunately, the reference of this paper, Murambiwa et. al. 2020 is missing in the bibliography.
The figures are clear and of good quality, each presenting important aspects of their findings. They shared their raw data for verification of results and points of discussion.

Experimental design

Research question was clearly stated and the gap in knowledge argued from literature. The design is appropriate to answer the research question.
The experimental design is straightforward and sound, to discriminate between treatment (co-infection) and non-treatment (Pb mono-infection) effects. The 42-day trial period allowed only 14 days period of co-infection, since Tz took 28 days to establish itself in muscle tissue. As much as the authors clearly state in the supplementary files and methods that day 0 is day 28 (when Pb co-infection took place, giving them 14 days to D42) I suggest in the annotated comments on the pdf, to always refer to the groups as mono-infection (Pb only) group and co-infection (Pb + Tz) group, the Tz only group poses no confusion.

A rigorous investigation was performed and high ethical and technical standards were upheld.

Validity of the findings

Their findings are corroborated by the results of this investigation and they took their liberty to use results from literature, which is actually research done in their laboratory. The effects of Pb and Tz co-infection on insulin levels, glycogen concentration and glucose concentration will still need further investigation, but these conclusions will form part of the basis or reference point for comparison. The experiments were appropriately controlled.

Additional comments

The paper is well written and the English is easy to understand. I suggested a few grammatical corrections on the pdf. If they can better distinguish the mono-infected and co-infected groups as stated under experimental design comments and on the pdf anotations, then the confusing element will be obviated. It was interesting to engage with the findings of this research.

---

## Round 0.2 · Minor Revisions

Although most of the changes suggested by the reviewers have been made, the manuscript still needs revision. The abstract must emphasize the importance and novelty of the study. Include a conclusion and delete excessive details related to the methods and results.

·

Basic reporting

Satisfactory

Experimental design

Comprehensive and adequate

Validity of the findings

All data were provided and valid

Additional comments

Overall, the queries have been answered.

---

## Round 0.3 · accepted · Accept

The abstract was changed accordingly.